# S100B Expression Plays a Crucial Role in Cytotoxicity, Reactive Oxygen Species Generation and Nitric Oxide Synthase Activation Induced by Amyloid β-Protein in an Astrocytoma Cell Line

**DOI:** 10.3390/ijms24065213

**Published:** 2023-03-08

**Authors:** Maria Elisabetta Clementi, Beatrice Sampaolese, Gabriele Di Sante, Francesco Ria, Rosa Di Liddo, Vincenzo Romano Spica, Fabrizio Michetti

**Affiliations:** 1Istituto di Scienze e Tecnologie Chimiche “Giulio Natta” (SCITEC-CNR), 00168 Rome, Italy; 2Department of Medicine and Surgery, Section of Human, Clinical and Forensic Anatomy, University of Perugia, 06132 Perugia, Italy; 3Department of Translational Medicine and Surgery, Section of General Pathology, Università Cattolica del Sacro Cuore, 00168 Rome, Italy; 4Department of Pharmaceutical and Pharmacological Sciences, University of Padova, 35131 Padova, Italy; 5Laboratory of Epidemiology and Biotechnologies, Department of Movement, Human and Health Scences, University of Rome “Foro Italico”, 00135 Rome, Italy; 6Department of Neuroscience, Università Cattolica del Sacro Cuore, 00168 Rome, Italy; 7IRCCS San Raffaele Scientific Institute, Università Vita-Salute San Raffaele, 20132 Milan, Italy; 8Department of Medicine, LUM University, 70010 Casamassima, Italy

**Keywords:** S100B, amyloid beta peptide, astrocytes

## Abstract

S100B is an astrocytic cytokine that has been shown to be involved in several neurodegenerative diseases. We used an astrocytoma cell line (U373 MG) silenced for S100B, and stimulated it with amyloid beta-peptide (Aβ) as a known paradigm factor for astrocyte activation, and showed that the ability of the cell (including the gene machinery) to express S100B is a prerequisite for inducing reactive astrocytic features, such as ROS generation, NOS activation and cytotoxicity. Our results showed that control astrocytoma cell line exhibited overexpression of S100B after Aβ treatment, and subsequently cytotoxicity, increased ROS generation and NOS activation. In contrast, cells silenced with S100B were essentially protected, consistently reducing cell death, significantly decreasing oxygen radical generation and nitric oxide synthase activity. The conclusive aim of the present study was to show a causative linkage between the cell expression of S100B and induction of astrocyte activation processes, such as cytotoxicity, ROS and NOS activation.

## 1. Introduction

Astrocytes are the first responders to noxious stimuli by undergoing cellular and functional transition, commonly referred as astrocyte reactivity, aimed to isolate damaged tissue and to enable the repair. The response comprises a change in astrocyte morphology and viability, and induction of inflammatory mediators, which may be either neurotoxic or neuroprotective. The molecular mechanisms inducing these alterations are not fully elucidated [1,2,3,4]. Among these different events, cytotoxicity, reactive oxygen species (ROS) generation and nitric oxide synthase (NOS) activation are widely regarded to constitute a significant part the phenomenon [5].

The S100B is an astrocytic protein, also currently regarded as a reliable biomarker of neural disorders, displaying characteristics shared with danger/damage-associated molecular pattern (DAMP) molecules, which actively participate in the tissue reaction to damage. In particular, intracellular levels of the protein, also when modulated in animal experimental models, have been shown to correlate with clinical symptoms and pathological/biomolecular parameters in various neural disorders [6,7,8]. Interestingly, S100B has also been shown to activate an inflammatory autocrine loop in astrocytes, dependent on its transmembrane immunoglobulin-like receptor for advanced glycation end-products (RAGE) and putatively involved in the propagation of reactive gliosis [9].

While it has been ascertained that S100B overexpression accompanies cytotoxicity, ROS and NOS activation, reasonably as a part of astrocyte activation processes [6,8], this study intends to demonstrate that the cell capability (including gene machinery) to express S100B is a needed prerequisite to induce the phenomena. For this purpose, this study uses an astrocyte cell line silenced for S100B and stimulated with amyloid beta (Aβ), as a paradigm factor known to stimulate astrocyte activation and correlative phenomena [10]. Thus, the conclusive aim of the present study was to show a causative linkage between the cell expression of S100B and the induction of astrocyte activation processes, such as cytotoxicity, ROS and NOS activation.

## 2. Results

First, a series of experiments were performed to evaluate the experimental system. Thus, we examined the concentration of Aβ most useful for standardizing our experiments. Therefore, we evaluated cell viability in the control cells in the presence of 5, 10 and 25 μM Aβ for 24 and 48 h (Figure 1). The most useful concentration for evaluating other experimental parameters resulted in being 10 μM, as also indicated by previous papers [11,12].

Next, S100B levels were evaluated in control and S100B-silenced cells after Aβ stimulation (10 μM). Aβ induced overexpression of S100B in control cells, as expected [6,8]. This effect was particularly evident after 48 h of stimulation (nearly 40%, *p* < 0.05). No changes in S100B levels were observed in silenced cells after Aβ treatment (Figure 2).

In order to dissect the possible role(s) of S100B on astrocyte machinery, cell viability was first analyzed. Figure 3 shows that while control cells exhibited a significantly reduced number of live cells after 24 and 48 h treatment with Aβ, with a percentage of dead cells at 20 and 35%, respectively, the silenced cells were consistently more resistant to cell death at the same time points.

Since it is known that Aβ also exhibits a toxic cellular effect through oxidative stress, which is regarded to constitute a crucial aspect of astrocyte activation [10], we assessed the ROS generation and the enzymatic activity of NOS in our experimental model. Figure 4A,B shows that Aβ that treatment at both 24 h and 48 h significantly increased both hydroxyl radicals and NOS activity in control cells. Specifically, the percentage of ROS increased to 150 and 200% after treatment for 24 and 48 h, respectively, while the effect on NOS activity was significant after 48 h of treatment, reaching almost double values (0.65 mU/mg protein) as compared to untreated cells (0.35 mU/mg protein). Silenced cells showed no significant changes in either ROS or NOS activity, indicating that S100B synthesis is needed for oxidative and inflammatory phenomena that might occur after Aβ stimulation.

## 3. Discussion

Astrocytes are the most widespread glial cells. In the case of AD pathogenesis, which directly involves Aβ, they both internalize and degrade Aβ and prevent extracellular plaque aggregation [10,13,14,15,16]. Activated astrocytes have been shown to markedly overexpress S100B, also during AD [8,17], and most of these astrocytes overexpressing S100B are closely associated with diffuse or neuritic Aβ plaques, thus being regarded as playing a significant role in pathogenic processes [18,19,20]. On the basis of these data, and of other coherent data obtained in other neural disorders [6,8], the notion that the S100B overexpression is strictly linked to astrocyte reaction may be taken for granted. Along this line, the present study offers information demonstrating that cell capability (including gene machinery) to express S100B is a needed prerequisite for the effect of an inflammatory inducers, such as Aβ, on cytotoxicity, ROS generation and NOS activation, which are also regarded as hallmarks of astrocyte activation [5], since in S100B-silenced cells Aβ stimulation was ineffective. This information unravels a crucial role of S100B in these phenomena, identifying a causative linkage between S100B expression and the occurrence of these phenomena participating in astrocyte reactivity. This may also be relevant in light of the heterogeneous cell population of astrocytes, which are known not to uniformly express the same molecular patterns [21,22].

In detail, our study showed that Aβ peptide raises intracellular levels of S100B and increases astrocyte cell death. Astrocyte activation is also known to induce an increase in oxygen radicals and NOS activation [10], which in fact is observed after Aβ treatment. These effects are abolished in S100B-silenced cells, where both survival and oxidative-inflammatory mechanisms were not affected after Aβ treatment. Interestingly, also in experimental animal models for multiple sclerosis, which are accompanied by an increase in oxidative–inflammatory mechanisms, these parameters were significantly reduced after treatment with drugs able to block the activity (pentamidine) or inhibit the astrocytic synthesis of S100B protein (arundic acid) [23,24]. It should also be noted that, in addition to multiple sclerosis, modulation of S100B has been shown to be relevant in the pathogenic processes of a number of neural disorders involving astrocytes, such as the aforementioned AD, Parkinson’s disease, amyotrophic lateral sclerosis, and acute traumatic and vascular neural injury [6,8]. Thus, the indication that S100B expression is a prerequisite for astrocyte activation processes may assume wide relevance.

Mechanisms accompanying S100B-induced effects on astrocytes are still undefined; however, it may be relevant to this study that the overexpression of S100B is regarded to act via the activation of nuclear factor NF-kB proinflammatory cascade [25,26], which is also known to be involved in astrocyte processes induced by Aβ peptide [10,27,28]. However, the cell mechanisms participating in these complex phenomena lie out the goal of this brief communication and might be the object of future studies. Reasonably, the present results are subject to criticism of the data obtained on cell lines, and cannot be uncritically applied to all in vivo astrocyte populations, which are known to be heterogeneous and subject to environmental interferences [21,22]. In this regard, since the issue of classifying and defining astrocytes based on structure, biomarkers and function has always been a matter of debate, it cannot be ruled out that the present results cannot be referred “tout court” to all types of astrocytes. However, in general terms, these results indicate that a downregulation of S100B in astrocytes activated by an inflammatory factor, such as extracellular Aβ, restricts astrocyte reactivity, including oxidative stress.

## 4. Materials and Methods

### 4.1. Preparation of Aβ Peptide

Aβ peptide was purchased from Sigma-Aldrich (Sigma-Aldrich, St Louis, MO, USA. Cat. Num. A9810). In accordance with previous papers, including our papers [11,12,29,30], to maintain Aβ in a monomeric state, we first dissolved the peptide in 1,1,3,3-hexafluoro-2-propanol (HFIP, Sigma-Aldrich, St. Louis, MO, USA), at a concentration of 1 mM. We intended to focus our study to one state of Aβ (monomeric), since the goal of our study was the reaction of astrocytes to an activating factor (in this case Aβ) in conditions when S100B is expressed or silenced, more than the comparative interaction of different (momoneric or oligomeric or polymeric) states of Aβ with the cell. After removal of the HFIP, the monomeric peptide was dissolved in DMSO at a final concentration of 2.5 mM, and stored at −80 °C until use (when it was diluted in PBS, according to the experimental design). In all control experiments, DMSO was added to the cell cultures at the same concentrations as the peptide solutions. Thawing and dilution to the target concentration in the appropriate culture medium was performed immediately before use.

Figure 1 shows that a 10 μM concentration offered a clearer effect on cell viability than 1 or 25 μM. Thus, we used this Aβ concentration (10 μM) for all experiments. In addition, this concentration also showed most clear results in previous in vitro studies [11,12].

### 4.2. Cellular Lines and S100B Silencing

U373 MG (human astrocytoma) cells were purchased from Sigma-Aldrich (Sigma-Aldrich, St. Louis, MO, USA. Cat. No. 08061901). The cells were cultured in Dulbecco’s modified eagle’s medium (DMEM, Invitrogen Co., Carlsbad, CA, USA), with the addition of 1% penicillin/streptomycin and 10% fetal bovine serum (FBS). They were then maintained at 37 °C in a humidified atmosphere with 5% CO_2_. The cells, once 60% confluence was reached, were transfected with stealth S100B siRNA oligo (Invitrogen—Waltham, MA, USA. Cat. Num. AM16708) or stealth-negative universal control (Invitrogen—Cat. Num. 12935300), using lipofectamine 2000 (Invitrogen—Cat. Num: 11668019), according to the manufacturer’s instructions. After 24 h, the cells were moved to serum-free medium and grown in this condition until use. Selective inhibition was confirmed after 2 weeks of culture, and viability during the same period was equal to that of control cells. In all experiments, the results obtained on S100B-silenced cells were compared to the negative control cells (U373 MG transfected with negative control siRNA).

### 4.3. Human S100B ELISA Assay

The S100B assay was determined in cell lysates and standards using the enzyme-linked immunosorbent assay (ELISA) kit, following the manufacturer’s protocols (human S100B ELISA Kit, Abcam, Cambridge, UK. Cat. Num. ab234573). We used the quantitative ELISA assay, which is highly standardized for S100B measurements, also in relationship to the wide clinical use, instead of other semi-quantitative assays, such as the Western blot assay, since we were interested in the final quantification of the protein and not in any possible characterization of the molecule. Briefly, the S100B standards (at a concentration of 0.63, 1.25, 2.5, 5, 10 and 20 ng/mL) or the samples of unknown concentration (50 µL of 0.5 mg protein/mL cell lysate) were added, in triplicate, to the wells pre-coated with the S100B antibody. The plate, after the addition of the antibody cocktail, was read at 450 nm with a microplate reader. The concentration of S100B in the samples was calculated in ng/mL by interpolating the absorbance values of the standard curve, and expressed as a percentage value compared to untreated control cells.

To prepare cell lysates, approximately 5,000,000 cells were used. Briefly, the cells were washed twice with cold PBS and then the pellets were lysed using the cell extraction buffer (Thermofisher—Waltham, MA, USA. Cat. Num. FNN0011). Then, the samples were centrifuged at 13,000 rpm for 10 min at 4 °C and the supernatants were collected. Protein concentration was determined using a protein assay (Bio-Rad, Hercules, CA, USA, Cat. Num. 5000002) in 96-well microplates, using a calibration line of BSA.

### 4.4. Direct Toxicity Study

For viability determination, cells were plated in 96-well plates at a density of 10,000 cells/well, and incubated for 24 h and 48 h in the absence (untreated) and presence of 10 μM Aβ. Cell viability was assessed using the MTS assay (CellTiter 96 Aqueous One Solution Cell Proliferation Assay, Promega, Madison, WI, USA. Cat. Num. G3582). The assay provided a measure of the normal metabolic state of the cells by recording the absorbance of each well with a microplate photometer (BioTek™ Elx800-Box 998; BioTek Instruments, Winooski, VT, USA) at a wavelength of 490 nm. Results are expressed as a percentage of cell viability compared to control cells (100%).

### 4.5. Quantification of Intracellular Levels of ROS

ROS generation was directly represented as a percentage relative to the control as the fluorescence intensity normalized to the number of cells.

Cells were plated in 96 black/clear-bottom wells (Greiner Bio-One, Kremsmünster, Austria) for the determination of reactive oxygen species. For detection after different treatments, a detector kit (DCFDA-Cellular ROS Detection Assay—Abcam, Cambridge, UK. Cat. Num. CB-P048-K) was used, initially non-fluorescent, which in the presence of ROS, is oxidized and its intensity, directly proportional to the amount of ROS, was measured in end-point mode at Ex/Em = 485/535 nm, using a BioTek Cytation cell imaging microplate reader (BioTek U.S., Winooski, VT, USA).

ROS generation was then represented as a percentage of the control as fluorescence intensity normalized to the number of cells.

### 4.6. NOS Activity

NOS enzyme activity was directly measured using the NOS Activity Assay Kit (BioVision, Kampenhout, Belgium. Cat. Num K2094), according to the manufacturer’s instructions. Briefly, 2 × 10^6^ cells were lysed with 300 µL of cold NOS lysis buffer (provided by the Kit) containing protease inhibitor cocktail, and centrifuged at 10,000× *g* at 4 °C for 10 min. On the collected supernatant, the protein concentration was calculated using the Bio-Rad Protein Assay (see above).

On a 96-well plate, 50 mL of sample (0.25 mg/mL protein) cell lysate and 5 µL NOS (positive control) were added. Subsequently, 40 μL of the reaction mixture (supplied by the kit) and 50 µL of the Griess reagent were added to each well. The optical density was measured at 540 nm with a microplate reader. A standard calibration solution (0, 250, 500, 750, 1000 pmol/well) of nitrite standard was used to generate the calibration curve. The activity of nitric oxide synthase was determined as B/TxC = mU/mg protein, where B is the amount of nitrite in the sample well from the standard curve, T is the reaction time (60 min) and C is the amount of protein.

### 4.7. Statistical Analysis

Every experiment was replicated at a minimum of three times and with at least seven replays per group. All results are displayed as the means ± SEM. Data were analyzed by a one-way ANOVA with the Newman–Keuls post hoc test, by using the Prism^TM^ 8:0 software (GraphPad, San Diego, CA, USA). Differences were considered significant at *p* < 0.05.

## Figures and Tables

**Figure 1 ijms-24-05213-f001:**
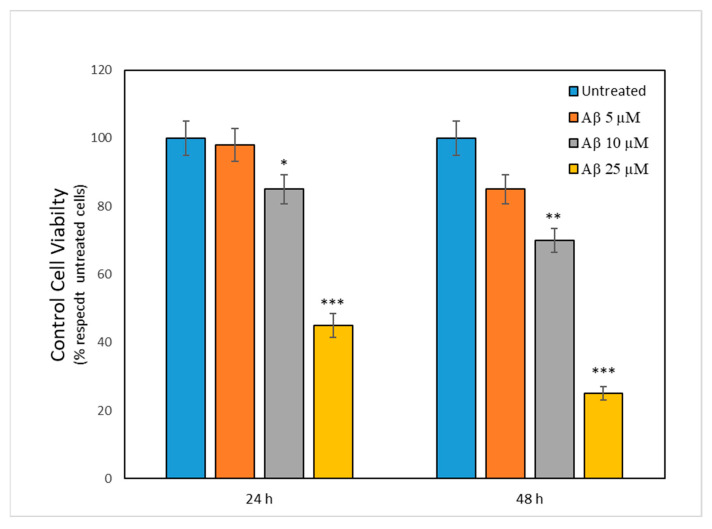
Viability for control cells in the presence of different concentrations of Aβ: 5 (orange bars), 10 (grey bars) and 25 (yellow bars) micromolar, after 24 h and 48 h of treatment. Viability is expressed as a percentage compared with untreated cells (100%—blue bars). Data are presented as the mean ± sd. Four replicates for each experimental group were considered. * *p* < 0.05, ** *p* < 0.01 *** *p* < 0.001 vs. control.

**Figure 2 ijms-24-05213-f002:**
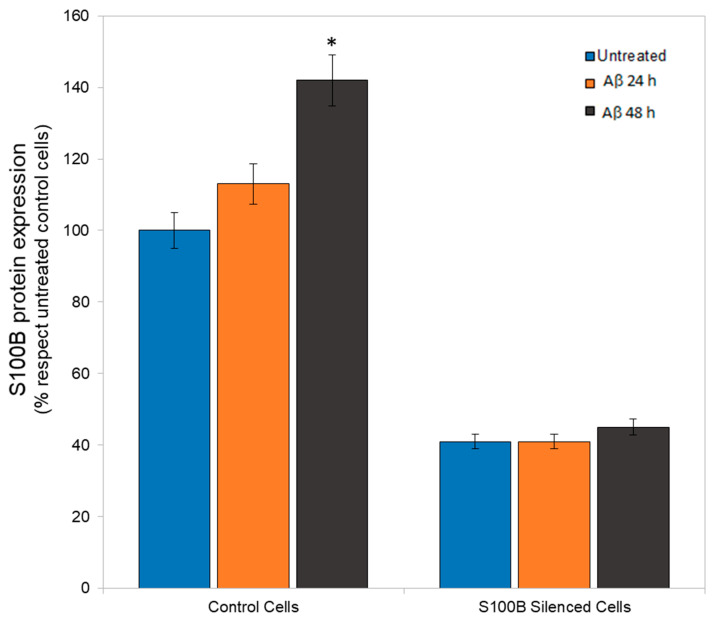
S100B protein levels in control and S100B-silenced cells, after treatment with 10 μM Aβ for 24 h (orange bars) and 48 h (black bars), or in the absence of Aβ (blue bars). Percentage values were obtained as indicated in the Materials and Methods section, and expressed as % in respect to the untreated control cells (100%). * *p* < 0.05 treated vs. untreated. The values of S100B expressed as ng/mL (respectively for control and silenced cells) were 10.41 vs. 3.8 (untreated), 11.8 vs. 3.7, (Aβ 24 h) and 15.00 vs. 3.9 (Aβ 48 h).

**Figure 3 ijms-24-05213-f003:**
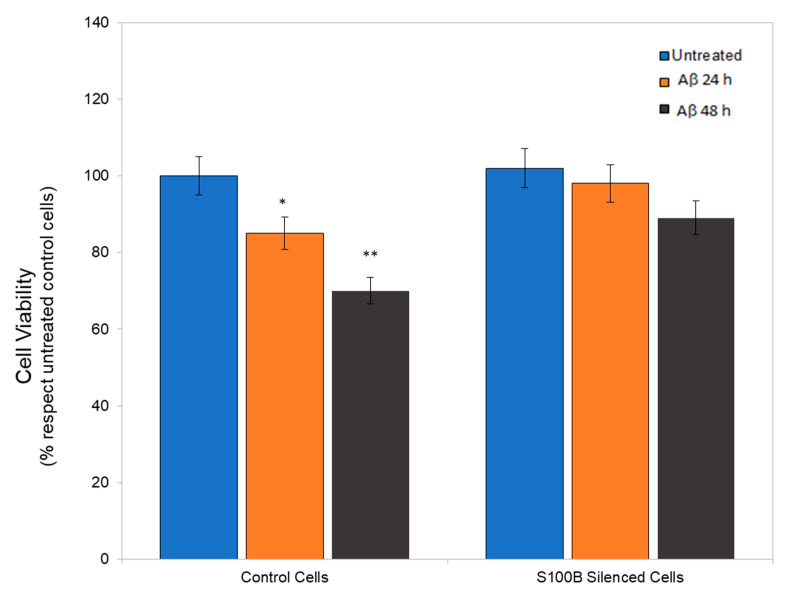
Cell viability for control and S100B-silenced cells after treatment with 10 μM Aβ, for 24 h (orange bars) and 48 h (black bars), or in the absence of Aβ (blue bars). Data are expressed as a percentage relative to the untreated cells (100%). * *p* < 0.05 and ** *p* < 0.001 vs. untreated.

**Figure 4 ijms-24-05213-f004:**
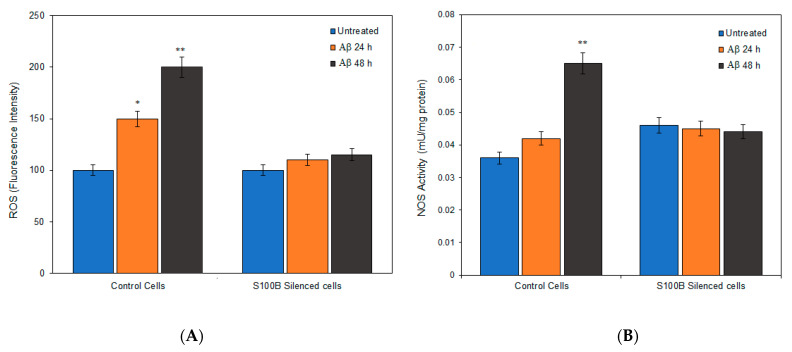
(**A**) ROS expressed as intensity fluorescence (percentage relative to the untreated cells) and (**B**) NOS activity (displayed as milliunits/milligrams of protein) for untreated cells (blue bars), treated with 10 μM Aβ for 24 h (orange bars) and 48 h (black bars). * *p* < 0.05 and ** *p* < 0.001 vs. untreated.

## Data Availability

Not applicable.

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
