# Peer review of "S100B Expression Plays a Crucial Role in Cytotoxicity, Reactive Oxygen Species Generation and Nitric Oxide Synthase Activation Induced by Amyloid β-Protein in an Astrocytoma Cell Line"

_ijms, 2023, doi:10.3390/ijms24065213_

Round 1

Reviewer 1 Report (New Reviewer)

The Authors present a brief communication on their preliminary results on the involvement of S100B in astrocytes inflammatory response after treatment with Ab.

S100B is a protein of  S-100 protein family, involved in the calcium homeostasis regulation. So I suggest for future experiments the analysis of intracellular Calcium levels. 

Minor: 

lane 173-174 the authors reported "  cell viability experiments, which were also performed using 1 uM and 25 uM  173  Ab concentrations, for 24h and 48h, and exhibited the clearest results at a concentration of  174  10 uM (Fig 1)." But I don't find it. 

Author Response

Reviewer 2 Report (New Reviewer)

Review of the manuscript entitled: S100B expression plays a crucial role in cytotoxicity, reactive oxygen species generation, nitric oxide synthase activation induced by Amyloid β protein in an astrocytoma cell line. The manuscript is interesting but many corrections need to be made. In both "summary" and "introduction" the aim of the manuscript is missing. Add for example “The aim of the present study was to…”. Moreover, “introduction” should end with aim of the manuscript. In the methodology, catalog numbers of key reagents should be provided, especially the siRNA, ELISA kits and so on.

If possible, the Authors should work on more professional figures. Figures should be more communicative, easy to understand without legend (of course the legend is important ;) ).

Key question. Whether the control cells were silenced? Control cells had added random siRNA because the control must contain them. Only this ensures the correct interpretation of the results. The experiment should look like this: random siRNA cells (negative control) and siRNA against S100B cells. In both groups, AB should be given to compare them.

Round 2

Reviewer 2 Report (New Reviewer)

Thanks for making the changes :)

This manuscript is a resubmission of an earlier submission. The following is a list of the peer review reports and author responses from that submission.

Round 1

Reviewer 1 Report

Comments

The present research article by Clementi et al. entitled “S100B expression plays a crucial role in astrocyte activation induced by Amyloid b protein in an astrocytoma cell line” demonstrates that Aβ treatment increases S100B protein expression in wild type U373 MG cells as compared to silenced cells and Aβ induced overexpression of S100B is significantly cytotoxic for U373 MG cells possibly due to increased ROS/NOS production. Authors proposed that S100B could be implicated as therapeutic target for controlling astrocyte activation in neurodegenerative diseases such as AD. There are few queries that need to be addressed as listed below:

Major queries/comments:

#1. In Fig.1, authors used ELISA to detect S100B protein expression in wild type vs silenced U373 MG cells. Why authors did not use western blot analysis? Authors also need to demonstrate the RNA transcripts (by qPCR) to demonstrate any alterations due to Aβ treatment. Also, authors need to measure the selective inhibition of S100B (how much %age inhibition) in siRNA silencing method. What was the viability of silenced cells upon selective inhibition at 2 weeks? Authors used 10µM Aβ for treating the cells throughout the study, why did they choose this dose. Authors need to show dose response curve for Aβ.

#2. Did author use Aβ oligomers or monomers for treating U373 MG cells? It is very important to specify and compare both conformations of Aβ for membrane internalization properties in U373 MG cells. Do Aβ and S100B interacts directly? If so, authors need to demonstrate this interaction using protein biochemistry approach such as co-immunoprecipitation or immunocytochemistry.

Reviewer 2 Report

Reviewer comment

 The manuscript „ S100B expression plays a crucial role in astrocyte activation induced by Amyloid β protein in an astrocytoma cell line”. Authors:Clementi ME, Sampaolese B, Di Sante G, Ria F, Di Liddo R, Romano Spica V. and Michetti F.

The experiments was performed on U373 astrocytoma cells and astrocytoma cells with silenced S100B with siRNA. The manuscript describes the effect of Aβ on cytotoxicity, ROS generation and NOS activity in S100B silenced astocytoma cells.

 Major points and experiments to be done

1.         The experiment outcomes evaluate the viability of S100B silenced astocytoma cells on Aβ-mediated  toxicity  and oxidative stress.The term “active” , or “reactive” astrocytes relates a complex morphological changes, upregulation of many gene expression, as for GFAP, Vimentin, the ECM proteins, proinflammatory mediators, and release of bioactive compounds, especially those stimulated by reactive microglia (IL-1β, TNFα, and C1q), as well  reduced phagocytic activity in response to diverse signals, Aβ also. Oxidative stress, what was determine by the authors, can trigger apoptosis, NFkB upregulation and proinflammatory reaction in astrocytes, but those was not evaluated. At least “astrocyte activity” should be specified by authors, since astrocyte responses are so diverged.

2.         For silencing S100B, authors used S100b siRNA and a quantification of the silenced protein was done by ELISA. In the Method section, the total protein determination in  the samples, nor units for S100B concentration, nor the range of S100B curve for standard are not described.

3.    It has been documented already,  that S100B overexpression is linked with astrogliosis after the brain insult and the various kinase signalling  pathways are postulated, in between a contribution in stabilization of F-actin cytoskeleton in astrocytes. Moreover, numerous research in AD mouse model, as well in vitro studies reported a S100B sharing in neuroinflammatory response of astrocytes. Then, more reasonable it was to investigate a specific cellular mechanism in activated astrocytes e.g. by Aβ after S100B silencing, than ascertain that S100B participates in astrocyte activation at all.

4.    Since, the relative value of S100B is reported in the manuscript, a statement in the Abstract, that ”the wild type astrocytoma cell line showed  an  overexpression of S100B” is based on the findings of other researchers. Nor S100B gene expression, nor WB was performed in this cell line by authors.

5.    Finally, the conclusion is so general that it fits almost to any research on the S100B effects, not to mention that no mechanisms have been investigated, so the involvement of S100B in the "control of astrocyte activation" cannot be stated.

6.     The level of S100B should be done by WB.

7.    The ROS levels should be measured in a shorter time (within hours) and in a different time points.

8.     Authors report an increased NOS activity after 48h in the regular astrocytoma cells, that may also indicate culture stress. To confirm a NOS sharing in astrocyte activation after S100B silencing, the expression of iNOS and its activity should be determined within a much shorter time.

9.    To confirm that S100B is involved in astrocyte activation,  NFkB activation, or GFAP expression, or other markers of astrocyte activation should be determined.

10. The experiment design should be reconsider also to switch on a   normal human astrocytes line with S100B silencing?

Round 2

Reviewer 1 Report

Thank you for addressing the queries/comments. Since, authors are using cultured astrocytoma cell lines, protein lysate sample could be obtained easily for using in Western Blot which is a better semi-quantitative assessment for protein levels. Also, qPCR analysis would provide supporting data for transcripts. A dose response curve of A-beta should be included in the manuscript to validate the effective conc. in the present experimental settings. Use of only monomeric A-beta in this study is understandable, however comparative data of monomeric vs oligomeric A-beta is important to fulfill the study conclusion. 

Author Response

We have to stress a preliminary consideration: this manuscript is submitted as a Communuication of preliminary results, according to the  Instructions for Authors for the International Journal of Molecular Sciences.

This preliminary consideration remains at the basis of the following responses

Reviewer 1:

Since, authors are using cultured astrocytoma cell lines, protein lysate sample could be obtained easily for using in Western Blot which is a better semi-quantitative assessment for protein levels..

We preferred the widely accepted quantitative ELISA method for protein assessment to the Western Blot assay, which is essentially semi-quantitative as also considered by the Reviewer, on the basis of the goal of our experiment, which was the quantitative assessment of the protein – i.d. the final objective- in silenced cells. In this respect, while ELISA is known to be especially reliable for S100B measurements, we want also to indicate, as an example, some recent papers, which also used ELISA to assess definite proteins, including S100B, in an experimental ground (1,-3). By the way, it is well known – and citations are not needed- that ELISA method is widely preferred to measure S100B protein in a clinical ground. In addition, while in general terms reasonably protein lysate sample could be obtained easily for using in Western Blot, cell lines at present are no more in our hands, so that unfortunatately we are unable to perform this very simple assay

1.Meng Y, Hao Z, Zhang H, Bai P, Guo W, Tian X, Xu J lncRNA NEAT1/miR-495-3p regulates angiogenesis in burn sepsis through the TGF-β1 and SMAD signaling pathways..Immun Inflamm Dis. 2023 Jan;11(1):e758. doi: 10.1002/iid3.758

2.Yu Y, Wang R, Zhang H, Wang J. Circ_0044411 silencing protects infantile pneumonia from lipopolysaccharide-induced cell injury by sponging miR-141-3p to inhibit CCL16 expression.Int Immunopharmacol. 2023 Jan;114:109425. doi: 10.1016/j.intimp.2022.109425

3.Callai EMM, Zin LEF, Catarina LS, Ponzoni D, Gonçalves CAS, Vizuete AFK, Cougo MC, Boff J, Puricelli E, Fernandes EK, da Silva Torres IL, Quevedo AS.  Evaluation of the immediate effects of a single transcranial direct current stimulation session on astrocyte activation, inflammatory response, and pain threshold in naïve rats. Behav Brain Res. 2022 Jun 25;428:113880. doi: 10.1016/j.bbr.2022.113880.

Also, qPCR analysis would provide supporting data for transcripts

Reasonably, as proposed by the Reviewer, qPCR data would provide supoporting data for transcripts. However, our goals  were the final levels of the protein (Fig 1) as well as the levels of ROS generation and NOS activity (Fig 3 A,B), which are different from transcripts as evaluated using qPCR. We have  also  to underline that data useful to the demonstration of results at the basis of the present work (i.d the silencing of S100B protein in cell lines, and the behaviour of ROS generation and NOS activity as affected by S100B expression after stimulation) appear to have been clearly shown. In addition, as outlined in the response to point 1, at present cell lines are unavailable in our hands, so that we are unable to perform a very simple assay such as qPCR is.

A dose response curve of A-beta should be included in the manuscript to validate the effective conc. in the present experimental settings

 A dose-response curve has been included as supplementary data in the manuscript, according to the request of the Reviewer.

Use of only monomeric A-beta in this study is understandable, however comparative data of monomeric vs oligomeric A-beta is important to fulfill the study conclusion. 

The goal of our work was not the study of Ab activity, which was merely used as a paradigm of inflammatory agent (other molecules could be used to the same purpose), but the needed role of astrocytic S100B expression on ROS generation, NOS activity and cytotoxicity after stimulation. For this reason, a comparative study of monomeric versus oligomeric Ab, although per se interesting, would lie ouside the goal of this study, and, as a consequence, reasonably seems not significantly to fulfill  the conclusion of our study.

Reviewer 2 Report

To the authors.

I have not changed my opinion  about the scientific strength presented in the reviewed manuscript. In the experimental set, the S100B content was measured after silencing its expression in glioblastoma cells, being a checking of the effectiveness of the procedure, and the achievements of the research is a demonstration of a less Aβ toxicity and ROS generate, and increased general activity of NOS. Intracellularly, the calcium-sensor S100B  regulates a variety cellular activities and knockdown/silencing of this protein will interfere in the intracellular Ca2+ signals, in between, in the Aβ oligomer mediated an increase in Ca2+-dependent ROS generation by NADPH oxidase activation which has to result also in reduced the cell viability. 

The question remains: how will the absence/reduction of S100B, consequently deprivation of one of the Ca2+ sensing proteins, affect the Ca2+ waives in astrocytes, an intracellular calcium buffering and mitochondrial function. With regard to the S100B up-regulation in glioblastoma cells, how to differentiate populations of astrocytes activated by inflammatory factors or amyloid tangles from resting astrocytes for a selective S100B downregulation in the brain ?

Hence,  the conclusion which is still visible in the pdf file entitles ijms-2149190-peer-review-v2 send for reviewer, that the results "support S10B as a therapeutic target having a focus on astrocytes" is invalid.  What therapy?  The other sentence is in the Abstract enclosed to Review Report Form.  

The last sentence of the conclusion may sound: ” These results indicate that a downregulation of  S100B  in astrocytes activated by extracellular amyloid oligomers, or tangles in AD may restrict astrocyte reactivity caused by oxidative stress”.

Minor comments.

1.      Change the last sentence of conclusions in the Abstract be present in the pdf file of the manuscript.

2.      Correct please the text edition (comas, dots and spaces) in the sections:

Abstract: line 26

Discussion:  lines 98, 117, 139, (NF)-kB exchange on NF-kB

Methods: lines  155, 166, 174,                                      

Author Response

We have to stress a preliminary cionsideration: this manuscript is submitted as a Communuication of preliminary results, according to the  Instructions for Authors for the International Journal of Molecular Sciences.

This preliminary consideration remains at the basis of the following responses

Reviewer 2.

The question remains: how will the absence/reduction of S100B, consequently deprivation of one of the Ca2+ sensing proteins, affect the Ca2+ waives in astrocytes, an intracellular calcium buffering and mitochondrial function. With regard to the S100B up-regulation in glioblastoma cells, how to differentiate populations of astrocytes activated by inflammatory factors or amyloid tangles from resting astrocytes for a selective S100B downregulation in the brain ?

The questions addresses now two novel topics not addressed in the first round of comments to the manuscript. In this respect, it should be noted that additional specific topics regarding action(s) of astrocytic S100B are potentially  very numerous, if not infinite.

In any case,  as indicated in the Discussion section of the manuscript, also in the light of the preliminary cionsideration specifying that this manuscript is submitted as a communication of preliminary results, the mechanism(s) of action of S100B in astrocytic response(s), even those possibly invoving Ca2+ waives and mitochondrial function, lie out of the goal of this communication .It should be noted, this respect,that the mechanism(s) of action of S100B (and similar molecules) in cell  responses in fact resulted to be very complex ( for reviews 1-3), and additional  studies appear still to be needed in order to obtain  conclusive information.

Likewise, again having in mind the preliminary consideration specifying that this manuscript is submitted as a communication of preliminary results, the topic of astrocyte heterogeneirty is ancient and wide, and reasonably might be opposed to all results concerning these cells. In any case,  considerations regarding this topic, while are already  reported in the Discussion section , have been enlarged in the present version of the revised manuscript

1.Michetti F, D'Ambrosi N, Toesca A, Puglisi MA, Serrano A, Marchese E, Corvino V, Geloso MC
The S100B story: from biomarker to active factor in neural injury..J Neurochem. 2019 Jan;148(2):168-187. doi: 10.1111/jnc.14574.

2.Michetti F, Di Sante G, Clementi ME, Sampaolese B, Casalbore P, Volonté C, Romano Spica V, Parnigotto PP, Di Liddo R, Amadio S, Ria F Growing role of S100B protein as a putative therapeutic target for neurological- and nonneurological-disorders..Neurosci Biobehav Rev. 2021 Aug;127:446-458. doi: 10.1016/j.neubiorev.2021.04.035

3.Balança B, Desmurs L, Grelier J, Perret-Liaudet A, Lukaszewicz AC DAMPs and RAGE Pathophysiology at the Acute Phase of Brain Injury: An Overview..Int J Mol Sci. 2021 Feb 28;22(5):2439. doi: 10.3390/ijms22052439

the conclusion which is still visible in the pdf file entitles ijms-2149190-peer-review-v2 send for reviewer, that the results "support S10B as a therapeutic target having a focus on astrocytes" is invalid.  What therapy?  The other sentence is in the Abstract enclosed to Review Report Form

The topic concerning drugs counteracting S100B, including drugs blocking S100B activity, such as pentamidine, or inhibiting S100B synthesis such arundic acid (ONO-2506), or neutralizing anti S100B antibodies, as also indicated in the Discussion section, is currently widely addressed (for reviews,1,2). The additional cited references merely indicate some recent papers published on this topic (3,4).

It may also be relevant in this respect that pentamidine is a drug already approved in human use, therefore ready for rapid transfer to clinical use in other disorders, potentially involving astrocytic S100B, as a drug repurposing..

Similarly, the injectable formulation of arundic acid (Proglia®) has completed phase III trials for acute-phase cerebral infarction, and the oral formulation (Cereact®) reached phase II trials in the UK for Alzheimer's disease and Parkinson's disease , which, such as cerebral infarction, also have been shown to involve S100B in pathogenic processes (for reviews,1,2).

In any case, the sentence cited by the Reviewer has been modified in the revised version of the manuscript.

1.Michetti F, D'Ambrosi N, Toesca A, Puglisi MA, Serrano A, Marchese E, Corvino V, Geloso MC
The S100B story: from biomarker to active factor in neural injury..J Neurochem. 2019 Jan;148(2):168-187. doi: 10.1111/jnc.14574.

2.Michetti F, Di Sante G, Clementi ME, Sampaolese B, Casalbore P, Volonté C, Romano Spica V, Parnigotto PP, Di Liddo R, Amadio S, Ria F Growing role of S100B protein as a putative therapeutic target for neurological- and nonneurological-disorders..Neurosci Biobehav Rev. 2021 Aug;127:446-458. doi: 10.1016/j.neubiorev.2021.04.035

3.Vizuete AFK, Leal MB, Moreira AP, Seady M, Taday J, Gonçalves CA.
Arundic acid (ONO-2506) downregulates neuroinflammation and astrocyte dysfunction after status epilepticus in young rats induced by Li-pilocarpine. Prog Neuropsychopharmacol Biol Psychiatry. 2022 Dec 21;123:110704. doi: 10.1016/j.pnpbp.2022.110704

4.Rinaldi F, Seguella L, Gigli S, Hanieh PN, Del Favero E, Cantù L, Pesce M, Sarnelli G, Marianecci C, Esposito G, Carafa M. inPentasomes: An innovative nose-to-brain pentamidine delivery blunts MPTP parkinsonism in mice.J Control Release. 2019 Jan 28;294:17-26. doi:10.1016/j.jconrel.2018.12.007.

The last sentence of the conclusion may sound: ” These results indicate that a downregulation of  S100B  in astrocytes activated by extracellular amyloid oligomers, or tangles in AD may restrict astrocyte reactivity caused by oxidative stress”.

We essentially modified the last sentence of the conclusion accordingly

Minor comments

All modifications requested in the Minor comments have been fulfilled in the present revised version of the manuscript

Round 3

Reviewer 1 Report

Thanks for your response to my queries, however I afraid that without including additional data on western blots, qPCR analysis and monomer treatments, the results drawn from this study are inconclusive.